# Application of Microfluidic Chips in the Detection of Airborne Microorganisms

**DOI:** 10.3390/mi13101576

**Published:** 2022-09-22

**Authors:** Jinpei Wang, Lixia Yang, Hanghui Wang, Lin Wang

**Affiliations:** 1College of Medicine, Xi’an International University, Xi’an 710077, China; 2Engineering Research Center of Personalized Anti-Aging Health Product Development and Transformation, Universities of Shaanxi Province, Xi’an 710077, China; 3Applied Research Center for Life Science, Xi’an International University, Xi’an 710077, China; 4Xi’an International Medical Center Hospital, Xi’an 710100, China

**Keywords:** airborne microorganisms, microfluidic chips, analysis and detection, bacteria, viruses, fungi

## Abstract

The spread of microorganisms in the air, especially pathogenic microorganisms, seriously affects people’s normal life. Therefore, the analysis and detection of airborne microorganisms is of great importance in environmental detection, disease prevention and biosafety. As an emerging technology with the advantages of integration, miniaturization and high efficiency, microfluidic chips are widely used in the detection of microorganisms in the environment, bringing development vitality to the detection of airborne microorganisms, and they have become a research highlight in the prevention and control of infectious diseases. Microfluidic chips can be used for the detection and analysis of bacteria, viruses and fungi in the air, mainly for the detection of *Escherichia coli*, *Staphylococcus aureus*, H1N1 virus, SARS-CoV-2 virus, *Aspergillus niger*, etc. The high sensitivity has great potential in practical detection. Here, we summarize the advances in the collection and detection of airborne microorganisms by microfluidic chips. The challenges and trends for the detection of airborne microorganisms by microfluidic chips was also discussed. These will support the role of microfluidic chips in the prevention and control of air pollution and major outbreaks.

## 1. Introduction

Microorganisms, as the most numerous, widely distributed and most complex biological groups in nature, play a vital role in the circulation of environmental substances, maintaining ecological balance and pollutant degradation. In recent years, the frequent outbreak of infectious diseases caused by pathogenic microorganisms in the environment has attracted people’s attention. The analysis and detection of microorganisms and biological macromolecules in the environment is an important research topic in environmental science, which helps to understand their migration and transformation laws and their impact on human society. The real-time analysis and detection of airborne microorganisms can achieve early warning and timely intervention to cut off the path of infection, so it is a significant development direction in the current environmental science research. Environmental microorganisms species include bacteria, actinomyces, fungi, viruses, and chlamydia [1,2]. Microbial aerosols contain various pathogenic and non-pathogenic components of biological origin, such as: active or inactive bacteria, bacterial or fungal toxins, viruses, macromolecular allergens, pollen, etc. [3]. The pathogenic components in the bioaerosol are mainly active viruses, active or dormant bacteria and fungi (spores). The particle size of these components is usually small. The particle size of the virus can be less than 100 nm, and the particle size of the bacteria is mainly 0.25~10 μm. The fungal spore size is mainly 1~30 μm [4]. Microbial aerosols are closely related to air pollution, environmental quality and human health. Although most microorganisms in the air are harmless, a small number of pathogens spread through the air and cause diseases, such as tuberculosis caused by *Mycobacterium tuberculosis* (TB), severe acute respiratory syndrome (SARS) and COVID-19 pneumonia caused by virus SARS-CoV-2, avian influenza caused by H1N1, H5N1 and H7N9 viruses, Ebola virus and pneumonia caused by fungi (*Aspergillus fumigatus*, etc.) [5,6]. These infectious diseases are relatively difficult to control or prevent; once an outbreak occurs, it easily causes not only widespread public panic but also serious economic losses.

Microbial aerosols can spread hundreds of meters or even thousands of meters through the air, which can spread to all corners of public places, greatly increasing the scope of their hazards [7]. Potential health hazards caused by pathogenic microorganisms in aerosols mainly depend on the pathogenicity of the microorganisms themselves; these pathogenic microorganisms can enter the human body in several ways: (1) direct source contact with aerosols through human mucous membranes or skin; (2) direct source contact with aerosols through hand-to-mouth contact, entering the digestive system; (3) through breathing into the respiratory system [8,9]. Bioaerosol transmission is the main mode of transmission for many diseases, including tuberculosis, severe acute respiratory syndrome (SARS), infectious diseases such as influenza, and respiratory diseases and even tumors [10]. Airborne microorganisms are also one of the important factors that cause nosocomial infections in hospitals. Nosocomial infection is considered to be the main obstacle to hospitalization and complications in adults and children, and about 10% of patients will be affected by nosocomial infection [11]. Nosocomial pathogenic fungi such as *Candida* and *Aspergillus fumigatus* can cause pneumonia. Pneumonia infection is uncommon in immunocompetent patients but can cause death in organ transplant recipients or immunocompromised patients [11,12]. Other pathogens in hospitals such as *Escherichia coli*, *Staphylococcus aureus*, respiratory syncytial virus, influenza virus, SARS-CoV-2, etc. are also threatening patients at all times [12,13,14]. Protective equipment removal and patient rooms had high concentrations per titer of SARS-CoV-2 (varying from 0.9 × 10^3^ to 40 × 10^3^ copies/m^3^ and 3.8 × 10^3^ to 7.2 × 10^3^ TCID50/m^3^) in hospital settings [15]. Therefore, hospitals have stricter requirements for microorganisms in indoor environments compared to ordinary cases. For low special microbial safety levels, the microbial testing and disinfection of medical equipment and medical environments should be carried out on a regular basis so that the level of pathogenic microorganisms in hospitals is maintained below the safety line to ensure patient health.

From this point of view, the rapid and efficient detection of airborne microorganisms is extremely urgent, especially the development of a technical method that can accurately and quickly detect on-site. As a result, a portable, rapid and high-throughput detection method is needed for the detection and analysis of environmental microorganisms. With the development of emerging technologies such as microfluidic chip technology, it provides opportunities for the rapid, efficient, real-time and on-site analysis and detection of airborne microorganisms. Compared with the traditional analysis methods, such as the culture, microscopy and counting methods, it has obvious innovation advantages. To detect airborne microorganisms, the usual practice is to first collect bioaerosols from the air and then detect pathogenic microorganisms. In recent years, several equipment systems based on microfluidics have been developed and reported for the collection and detection of microorganisms. Therefore, this paper summarizes the detection and analysis of the microfluidic detection of bacteria, viruses and fungi in the air in order to provide help for the subsequent development of airborne microbial detection and to provide support for the prevention and control of the continuous outbreak of major epidemics such as COVID-19.

## 2. Microfluidic Chips

A microfluidic chip, also known as a lab-on-a-chip, can complete multiple steps of sampling, dilution, reaction, separation and detection on a chip and use microfluidics to complete the analysis and detection device [16]. Compared with other detection methods, microfluidics technology has the advantages of a simple operation, short detection time, good selectivity, high resolution, low reagent consumption, low cost, good portability and high-throughput detection. It has been applied in real life in some fields [17,18,19]. With the development of information technology, the combination of microfluidic chips and various information technologies can realize the direct readout of detection data, and it is expected to be directly applied to people’s lives to monitor environmental changes.

The commonly used materials for microfluidic chips are monocrystalline silicon wafers, quartz, glass and organic polymers such as polymethylmethacrylate (PMMA), polycarbonate (PC) and polydimethylsiloxane (PDMS), etc. and merging materials such as paper-based, polylactic acid, hydrogel, etc. [20,21,22]. The basic processing technologies of existing microfluidic chips mainly include injection molding, photolithography, etching, hot pressing, molding, manual cutting, 3D printing, etc. [23]. At present, the commonly used detection methods are mainly optical detection, electrochemical detection and mass spectrometry detection [24]. In addition, surface-enhanced Raman scattering detection (SERS), surface plasmon resonance spectroscopy (SPR), colorimetric detection, loop-mediated isothermal amplification (LAMP), immunosensors and other detection technologies are gradually used in combination with microfluidic chips. Microfluidic chips can integrate various functions for the high-throughput and automated analysis of microbes, as illustrated in Figure 1.

### 2.1. Materials for the Fabrication of Microfluidic Chips

Due to the good chemical inertness and thermal stability of silicon materials, high-precision two-dimensional patterns or three-dimensional structures can be reproduced by etching and photolithography. However, silicon materials are easily damaged, have a high cost, have a poor insulation and light transmittance and have complex surface chemical behaviors. These shortcomings limit its application. The optical properties of quartz and glass are excellent, and microstructures can be engraved on quartz and glass using lithography and etching techniques similar to those of silicon. Moreover, these two materials have surface adsorption and reaction capabilities that are beneficial to surface modification, so quartz and glass have become commonly used materials for making microfluidic chips [26,27]. However, they are relatively expensive, especially quartz. The properties of polydimethylsiloxane (PDMS) are completely different from those of silicon materials. It is a transparent soft elastic material. This polymer was first used in the fabrication of chips by Kumar and others at Harvard University in 1993 [28]. Due to the elastic properties of PDMS, micro-valves of various structures can be fabricated on the chip. The gas permeability of PDMS enables cells and various microorganisms to be cultured in the PDMS chip. The polymer is non-toxic, inexpensive and simple to manufacture, and it has become a material widely used in the manufacture of chips [29,30,31,32]. However, its surface is highly hydrophobic, and it is easy to adsorb some hydrophobic samples to be tested, or, bubbles are easily generated in the channel during the fabrication process [33,34]. Polymethyl methacrylate (PMMA) is also a polymer material commonly used in chip fabrication [35,36]. It has the advantages of a low cost, a good light transmittance and good electrical properties, and it can form a relatively stable material under the action of an electric field. The electroosmotic flow and PMMA are easy to process and shape [37]. COC (copolymer of cycloolefin) is a thermoplastic amorphous polymer with a cyclic olefin structure [38]. It not only has optical properties comparable to PMMA and higher heat resistance than polycarbonate (PC), but also has dimensional stability, low fluorescence background of COC, easy processing, acid and alkali resistance and good biocompatibility characteristics, so it has a wide range of application prospects [39]. In 2017, Guo et al. [40] applied ionomer-enhanced electrophoresis technology to COC chips and used it to determine rhodamine dyes. In 2018, Li et al. [41] applied the COC chip to the detection of HIV p24 antigens based on smartphones.

Microfluidic paper-based analytical devices (μPADs) are a new concept first proposed in 2007; they are a novel microfluidic technology using filter paper as a test carrier [42]. Compared with materials such as silicon and glass, paper chips are cheaper and have the advantages of miniaturization, easy storage and transportation, a low detection background, good biocompatibility and the ability to immobilize biological macromolecules [42,43,44,45,46]. Microfluidic Paper-based Analytical Devices can also make use of composite materials [47], valve technology [48], lightbox and cellphone technology [49] and optimized device architecture [50] to enhance detection performance. In recent years, paper chips have been widely used in food safety, environmental residual pollution, pathogen detection and other fields [51]. Research on paper chips in infectious disease and pathogen detection shows priority, and paper chips are well suited for the point-of-care detection of viruses, especially in poor areas [52]. However, the sample easily evaporates during transportation, and it is easy to remain in the chip channel. For some samples with low surface tension, it is easy to leak in the chips. These shortcomings will reduce the sample utilization rate.

Polylactic acid (PLA) is a common biodegradable thermoplastic made from natural material lactic acid, and lactic acid is made by the simultaneous saccharification and fermentation of starch, so PLA has a good biocompatibility and has been adopted in the biomedical arena [53]. Approved by the Food and Drug Administration for its application to the human body [54], PLA is more environmentally friendly and degradable than other plastics [55]. PLA has become a material for fused deposition modeling (FDM) due to its low elasticity, low melting and good biocompatibility, and it is mostly used in 3D printing and microfluidic devices [56]. Although PLA has a wide spectrum of applications, there are certain limitations such as the slow degradation rate, hydrophobicity and low impact toughness associated with its use [57].

Hydrogels have become a new type of material for fabricating microfluidic devices in recent years because of their good permeability and biocompatibility [58,59]. In 2005, Mario’s research group first proposed the use of hydrogels to make microfluidic devices [60], and in 2007, the research group proposed a lithography technique to construct functionalized microfluidic structures in calcium alginate hydrogels [21]. Microfluidic tissue scaffolds were fabricated using hydrogels. In this way, convective mass transfer is used to control the distribution of soluble chemicals within the scaffold and can be used for complex engineered tissues.

### 2.2. Methods for the Fabrication of Microfluidic Chips

Microfluidic chips use specific microfabrication techniques to produce micron-scale channels and other components on the chip material substrate. The techniques mainly include photolithography and etching, which can engrave various images on the chip with high precision [26]. Photolithography is a process of forming desired images on materials such as silicon and glass using photoimaging and photosensitive adhesive technology, which generally includes pretreatment, gluing, prebaking, exposure, development and modeling. In 2019, Wlodarczyk et al. [61] proposed a new technology for making glass chips. In the process of making microfluidic devices, only a picosecond pulsed laser system is needed, and the picosecond pulsed laser system is used to directly generate microfluidics on glass. Control the pattern, drill in/out on the other piece of glass and glue the two glass sheets together to facilitate the encapsulation of the pattern generated by the laser system from the top. This technique enables the fabrication of a fully functional microfluidic device within a few hours and does not require any masks, hazardous chemicals and other expensive large-scale tools. Because of their elasticity, polymer microfluidic chips are mainly fabricated by the hot pressing method, molding method, laser ablation method and soft etching method.

(1) The injection molding method is to use photolithography and etching to make a male mold with a raised microchannel structure on glass or silicon materials and then heat the raw material in the injection to make the raw material become fluid and press into the mold. After cooling down to room temperature, the polymer chip with a microchannel structure can be obtained by peeling it off from the mold [62]. The commonly used molds are mainly ring SU-8 negative light glue or positive glue and can also be made of rubber and other materials. The injection molding method is simple and convenient, and the used mold can be reused. It has been used in the fabrication of chips made of cyclic olefin copolymer (COC) [63], PDMS [33] and PMMA [64,65].

(2) The laser ablation method is a non-contact processing technology that uses a mask directly according to the computer design to control the position of the XY direction of the laser and uses high-energy laser irradiation on the material to destroy the interaction between polymer molecules so as to make different types of microstructures and channels [66]. In 2002, Henning et al. [67] first applied CO_2_-laser micromachining technology to produce microfluidic systems made of PMMA materials. In 2018, Ongaro [55] applied this technology to the fabrication of PLA chips, used the SLAM (simultaneous localization and mapping) method to eliminate the recast layer on the cut material, further improving the processing performance of a CO_2_-laser, and demonstrated a five-layer microfluid fabricated with it. The advantages of this method are that the resulting microstructure is less damaged, the quality depends on the material used, the flexibility is high and the adsorption is low, but the production efficiency is low and the laser used is expensive. These inadequacies limit the further development of this technology.

(3) Soft lithography. Younan et al. [68] replaced the hard mold with an elastic model to make chips, which is called soft lithography. This technique is more flexible, has no precision limitations and uses equipment that is less expensive and is easier to operate, but the soft model used is too soft to achieve large aspect ratios.

(4) A hot-press molding method is a method in which a micro-structured mold is covered on a polymer substrate, and it is heated and pressurized to obtain a micro-structured chip. It was first proposed by MacCrehan et al. in 2000. They made a PMMA microfluidic chip with microchannels using chrome-nickel alloy wire and a silicon male mold with microchannels as molds [29]. In 2004, Yang et al. [63] used single-crystal silicon as a positive mold to make a COC chip and combined it with electrospray mass spectrometry. Wei [69] used an 80 μm copper wire as a positive mold, heated it at 127 °C for 25 min to press the copper wire into the COC chip and finally corroded the copper wire with concentrated nitric acid to obtain a COC chip with microchannels.

(5) 3D printing technology is based on digital model files, using PLA material to rapidly form the designed pattern through layer-by-layer printing. Since this method is flexible in design, does not require post-processing, can be used for surface and local processing, electrode and membrane integration and other advantages, it is applied to the production of microfluidic devices. In 2016, Gong et al. [70] applied 3D printing technology to the fabrication of microfluidic valves, pumps and multiplexers. However, there are still some challenges in 3D printing, such as the long printing time required to make complex structures, the complicated material fabrication and the limited resolution [71].

## 3. Collection and Detection of Airborne Microorganisms

To study the transmission routes of bioaerosols, a common practice is to first collect the bioaerosols from the air and then detect the pathogenic microorganisms.

### 3.1. Collection of Airborne Microorganisms

In recent years, related technologies and methods such as the enrichment and detection of airborne microorganisms have received more and more attention. Airborne microbial sampling methods are diverse, and many are still in the developmental stage. To date, neither a single sampling method nor a single sampling standard protocol is available for the collection of various types of airborne microorganisms. The solid impact method, liquid impact method and filtration method are commonly used collection methods for airborne microorganisms. Besides these three methods, the natural sedimentation method, electrostatic deposition method and microfluidic chips method are sometimes used to collect airborne microorganisms.

(1) The solid impact method uses a sampler to collect bioaerosols from a solid medium (such as agar), which has the advantages of a low cost and easy handling. The sampler draws in the air aerosol and forces the gas to change direction, causing particles with a high inertia to strike the solid medium surface [72]. Microorganisms collected by the solid impact sampler can only be counted by the culture method; when sampling highly polluted air, the overlapping of colonies makes it difficult to count. (2) Liquid impact method: Normally, air is sucked in through a narrow inlet tube. Once the air impacts the surface of the medium liquid, the suspended particles will contact the liquid medium and be collected [73]. However, the liquid impactor is used before it is sterilized, and the evaporation of the liquid may cause the loss of microorganisms. (3) Filtration method: The filter membrane of the filter is generally a cellulose membrane, nylon membrane, carbonate membrane or glass fiber membrane with a pore structure [74,75]. The filter method sampler has a high capture efficiency for microorganisms whose size is larger than the micropore diameter of the filter membrane surface. (4) The natural sedimentation method is a simple and economical sampling method. Under certain conditions, when the medium is exposed to the air, microorganisms will be collected into the medium under the action of gravity [76]. The natural sedimentation method mainly collects macromolecules’ biological particles; the collection efficiency of small molecular biological particles is not high, so it cannot correctly indicate the microbial information in the air. (5) The microfluidic chip collection method adopts a unique fishbone or herringbone structure, which can disturb the air and cause confusion in the airflow, thereby adsorbing the microbial particles in it [77,78]. The advantage of microfluidic chips is a high enrichment efficiency, small elution volume and simple operation.

### 3.2. Detection of Airborne Microorganisms

The detection of microorganisms is the second largest step in bioaerosol monitoring. Microbial detection methods can be divided into two categories: the culture method and the non-culture method. The culture method is a traditional method for microbial detection which is low-cost simple to operate. Under appropriate culture conditions (including time, temperature, oxygen concentration, etc.), the collected microorganisms can form colonies (CFUs) on the culture medium. The identification techniques of cultured microorganisms mainly include: microscopy, laser-induced fluorescence (LIF), matrix-assisted laser desorption/ionization (MALDI) and laser-induced breakdown spectroscopy (LIBS) [79,80]. The disadvantage is that the proportion of microorganisms that can be cultured and identified in the environment is very small (about 10%), so the culture method cannot provide information about the total number of airborne microorganisms.

With the development of fluorescent dyes, it has become possible to quantify all microorganisms (including culturable and non-culturable microorganisms) collected in liquid culture media. The development of genomics and next-generation sequencing technologies helps not only to identify and quantify the microbial load but also to understand how microbial populations may change. In addition, chromatography, immunoassays and polymerase chain reaction (such as PCR) have helped to expand the identification of microorganisms. However, several disadvantages have greatly limited their practical application, including the consumption of large quantities of reagents requiring expensive equipment and trained technicians, along with the limited accuracy and stability [81,82,83]. Microfluidic technology has the advantages of a high efficiency, a high specificity, a high throughput, simplicity, rapidity and a small number of reaction reagents, and it has attracted more and more attention.

## 4. Detection of Airborne Microorganisms by Microfluidic Chips

At present, most of the analysis and detection methods of airborne microorganisms are to isolate microorganisms by sampler and plate culture method, and then analyze and detect by microscope, PCR amplification or gene chip technology [84,85]. However, this method is time-consuming and complicated to operate and has a low sensitivity. It usually takes 18–24 h and cannot meet the needs of point-of-care testing. Many pathogenic microorganisms can survive in the natural environment but cannot be cultured. The above-mentioned traditional methods are difficult to detect. All of these are not conducive to the early warning and prevention of pathogenic microorganisms. There are still many challenges in the rapid detection and accurate qualitative and quantitative analysis of airborne microorganisms [86]. The development of microfluidic chips meets the needs of scientific research and the market, combined with nucleic acid extraction, protein extraction, PCR amplification, DNA sequencing and immunoassay detection technologies, and they can be used for the detection of airborne microorganisms [87,88]. At present, most studies have focused on the microfluidic detection of microorganisms in liquids, and there are relatively few reports on the detection of airborne microorganisms.

### 4.1. The Role of Microfluidic Chips in the Separation of Microorganisms

In order to conduct the direct and real-time detection of microorganisms in the air, samples must be collected, but environmental particles such as dust are also trapped in these samples. Therefore, it is very important to separate target bacteria from non-biological particles or selectively collect microorganisms before detection. Moon et al. [89] proposed a method to separate microorganisms in the air, using a microfluidic device and dielectric electrophoresis (as shown in Figure 2A) to successfully isolate 90% of *Micrococcus luteum* from a mixture of bacteria and dust. Kang et al. [90] developed an air microbial detection chip, as shown in Figure 2D, which used an inertial impact system to separate microbes with an aerodynamic particle size and distinguished microbes and non-biological particles with a micro-fluorescence microscope. Meltzer et al. [91] used a microfluidic chip integrating two detection methods for the separation and detection of bacteria and viruses in the air, in which direct linear analysis (DLA) was used to detect bacteria and digital DNA was used to detect viruses, as shown in Figure 2B. Hong et al. [92] developed a microchannel-based aerosol separator that can separate virus particles and bacteria in the air by size, as shown in Figure 2C. Using a real-time aerosol-measuring instrument and the quantitative polymerase chain reaction (qPCR) analysis method to determine the separation ratio of each bioaerosol, *Staphylococcus epidermidis adenovirus* can be isolated. Choi et al. [93] reported a microfluidic sampler for air microbial enrichment and detection, which has a better collection efficiency for *Staphylococcus epidermidis*. The curved microchannel was used to collect bioaerosols into liquids. Due to the centrifugal and drag forces on the bioaerosols, the direction of bioaerosols was changed towards the liquid. The content of airborne microorganisms is low, and it is susceptible to particulate matter during the separation process. Therefore, a microfluidic chip is required to combine with other technologies such as flow cytometry, mass spectrometry and other instruments to collect and separate airborne microorganisms.

### 4.2. Application of Microfluidic Chips in Bacterial Detection

At present, the detection of bacteria in the air mainly focuses on the detection of some model bacteria, such as the detection of *Escherichia coli*, *Staphylococcus aureus* and *Pseudomonas aeruginosa*. Jing et al. [94] reported a microfluidic device (as shown in Figure 3A) that uses a staggered herringbone mixer (SHM) structure to quickly capture and enrich *E. coli* and *Mycobacterium smegmatis* in the air. The enrichment efficiency can approach 100% within 9 min. Bian et al. [95] reported a portable bioaerosol sampler based on a microfluidic chip (as shown in Figure 3C), which has a sampling efficiency of 99.9% for *Vibrio parahaemolyticus.* Airborne *Mycobacterium tuberculosis* is the main source of infection of tuberculosis. Then, Jing et al. [77] designed a microfluidic chip for the detection of *Mycobacterium tuberculosis*, as shown in Figure 3D. Due to the staggered herringbone structures, the chaotic flow was induced to create more chances for the collision between the bacteria and the inside wall of channels, resulting in a high capture efficiency (almost 100%) within 20 min. The entire detection time is about 50 min, which is much lower than the 29 days of conventional methods. Afterwards, Jiang et al. [78] combined a high-throughput flow PCR chip with an airborne microbial capture chip to detect six common bacteria (*Staphylococcus aureus*, *Klebsiella pneumoniae*, *Pseudomonas aeruginosa*, *Citrobacter kleinii*, *Enterococcus faecalis* and *E. coli*) for continuous flow analysis. Without the purification of DNA, the detection limit of *E. coli* reached approximately 118 cells. The amount of sample required for each set of chip PCR reaction is only about 0.13 uL, which is suitable for the analysis and research of rare and precious samples. Fuchiwaki et al. [96] proposed a new type of microfluidic system that uses vapor pressure to achieve polymerase chain reaction (PCR). The anthrax aerosol was sampled and detected, with a detection time of 8 min, as shown in Figure 2B. Liu et al. [97] reported a response system (as shown in Figure 3E) for the rapid detection of pathogens in the air. The reagents used for loop-mediated isothermal amplification (LAMP) are embedded in the reaction chamber of the chip with hydrogel, and the detection of *Pseudomonas aeruginosa* in the air can be completed within 70 min. The system is suitable for point-of-care applications. Choi et al. [98] developed a real-time surface-enhanced Raman spectroscopy (SERS) detection platform using a microfluidic chip, as shown in Figure 3F. The detection limit of *Staphylococcus epidermidis* in the air is about 10^2^ CFU/mL, which provides an excellent solution for the prevention of hospital air infections. In addition, Stachowiak et al. [99] proposed a system for the automatic detection of air bacteria based on microfluidic chips. The detection and analysis are mainly based on protein profile analysis. This system can detect the spores of *Bacillus subtilis*. Jia et al. [100] developed a low-cost paper-based microfluidic device that combines a drone aerial mobile sampling system with a smart phone colorimetric analysis to quantitatively detect viruses and bacteria in the air.

Microfluidic chips are involved in the detection of bacteria in sampling, enrichment and detection. However, the quantitative detection of bacteria in the air is still lacking. It needs to be further combined with sequencing technology and flow cytometry technology to improve its detection sensitivity.

### 4.3. Application of Microfluidic Chips in Virus Detection

The efficient and rapid collection of air samples is an important step in the analysis and detection of airborne microorganisms. Airborne viruses, such as H1N1 influenza, SARS, avian influenza and new coronaviruses, are highly transmissible and contagious among humans. Therefore, it is necessary to develop an airborne virus system that can be detected in real time, but this could be a long-term challenge in this field.

Influenza Aflu is a highly contagious and rapidly spreading airborne disease that requires near-real-time monitoring. However, influenza viruses are present at extremely low concentrations in aerosol samples and are susceptible to interference from dust particles. Shen [101] developed a real-time monitoring system (as shown in Figure 4A) for H3N2 influenza virus in the air based on a microfluidic chip. It took about 12 min from the release of the simulated aerosol to the detection, and the concentration of influenza Aflu H3N2 virus was detected to be 10^4^ viruses/L or lower. Kwon et al. [102] developed a microfluidic device, using a miniature spectrometer or a mobile phone camera as an optical detector, to detect and quantify the H1N1 virus in aerosol samples captured from a simulated classroom. The detection limit of the spectrometer for real air samples is 1 pg/mL, and the detection limit of the mobile phone camera for real air samples is 10 pg/mL, which is several orders of magnitude higher than that of other methods. The detection time is only 5 min and 30 s, which has high commercial potential.

The new coronavirus disease (COVID-19) pandemic has also been demonstrated to transmit through bioaerosols. Most of the general detection methods for new coronaviruses in the air involve the use of a special adsorption membrane to enrich the virus in an aerosol state and then dissolve it into a liquid state for routine nucleic acid detection. However, this method involves many steps, complicated operation, large size of the device and poor portability. With the outbreak of the 2019 novel coronavirus disease (COVID-19) pandemic, experts recommend the use of a point-of-care diagnostic system integrated with an air pathogen monitoring machine to prevent and control the early spread of β-coronavirus type 2 (SARS-CoV-2). In order to meet the needs of the rapid on-site collection and detection of SARS-CoV-2 virus, Xiong et al. [103] constructed a SARS-CoV-2 aerosol sampling system integrated with a small-volume rotating microfluidic fluorescent chip (as shown in Figure 4B). The system includes an automatic centrifugal sampling system, an injection system and a heating system for detection by enriching the virus and extracting viral nucleic acid. At present, 115 clinical samples have been successfully tested, and it was found that the system has 100% specificity, a high sensitivity (10 copies/μL) and a good detection accuracy for the rapid detection of SARS-CoV-2. The rapid on-site monitoring and diagnosis of nucleic acid provides a good monitoring and diagnosis tool for the follow-up epidemic prevention work. In addition, the “Air Virus Collector” and “Microfluidic Detection Chip” of Aigen Technology Co., Ltd. can be used to detect SARS-CoV-2 viruses in the air. They have obtained the European Union CE certification and can be used commercially. The supporting device mainly adopts a three-step method: air collection, nucleic acid extraction and spot detection, and the test results can be obtained within 30–60 min. The device can detect SARS-CoV-2 viruses in multiple scenarios, provide timely warnings and troubleshoot suspicious environments, but its portability needs to be improved. A paper-based microfluidic chip and immunofluorescence assay were proposed as one method to detect airborne SARS-CoV-2 [104].

At present, the microfluidic chips have a high sensitivity for detecting viruses in the air, but in terms of virus enrichment and confirmation of whether it is active, the experimental results are not ideal, and there are few practical applications.

### 4.4. Application of Microfluidic Chips in Fungi Detection

The airborne transmission of fungi, including *Aspergillus* species, is the main cause of human asthma. Monitoring the content of specific fungi in the air is one of the key technologies to prevent airborne diseases. Xu et al. [105] proposed a microfluidic chip method for accurately extracting pathogenic microorganisms directly from airflow (as shown in Figure 5A). The results showed that the extraction rates of mold spores and gray mold spores were 89% and 76%, and the removal rates were 98% and 87%, respectively. The high-purity and accurate extraction of fungal spores was achieved, which can be used for the early and rapid detection of fungal diseases. Li et al. [106] demonstrated an integrated microfluidic system capable of enrichment and high-throughput detection of fungal spores of *Aspergillus niger* in the air. The device, as shown in Figure 5B, is based on immunofluorescence analysis for the semi-quantitative detection of *Aspergillus niger* spores. From enrichment to fluorescence detection, the whole process can be completed within 2–3 h, and the detection limit was 20 spores. The microfluidic system has integrated sampling and analysis functions, avoiding additional sample concentration steps, and used a micro-fluorescence detector to read the detection results, showing the potential for the instant detection of other pathogenic fungal spores. In addition, some biomarkers, such as microbial metabolites, endotoxins and mycotoxins, can be detected by integrated microfluidic chips to determine the types of fungi in the air. Liu et al. [107] designed a rapid multiplex nucleic acid detection system for airborne fungi, which includes an integrated DNA release device and a portable microfluidic chip. The integrated DNA release device was 3D printed. The integrated DNA release device combined mechanical lysing and biochemical reagent treatment to automate DNA release. The microfluidic chip was capable of multiplex nucleic acid detection. The process took only 90 min from sample input to readout, and the detection limit of this system was 4 × 10^4^ spores per test, meeting the requirement of early warnings. Furthermore, the system was economical and user-friendly, without resource restrictions.

## 5. Challenges and Future Trends

The monitoring of airborne pathogenic microorganisms is very important for the prevention and control of infectious diseases. The collection, analysis and detection of biological aerosols is an important part of monitoring pathogenic microorganisms in the air. At present, many emerging methods and devices based on microfluidic chips have been used to collect and detect bioaerosols. However, there are still some shortcomings and areas that need to be improved.

(1) The concentration of pathogenic microorganisms in the air is usually very low, so the collected bioaerosol is usually very small, and it is difficult to directly use it for detection due to the limited sensitivity of existing detection methods. Therefore, improvements in bioaerosol concentration and signal amplification are still needed.

(2) There are both live and dead pathogenic microorganisms in bioaerosols. However, most of the current detection methods do not have the ability to distinguish the biological activity of microorganisms, resulting in the inaccurate detection of pathogenic microorganisms in bioaerosols. Furthermore, many enrichment methods may have an impact on the biological activity of live microorganisms, leading to an underestimation of live pathogenic microorganisms in bioaerosols.

(3) Currently, ubiquitous bioaerosol monitoring consists of two separate components: bioaerosol collection and pathogen detection. Integrating bioaerosol collection and detection into a single unit is still a big challenge. Continuous and automated bioaerosol monitoring is urgently needed for on-site collection and detection.

(4) Although microfluidic chips can greatly reduce the consumption of samples and reagents, the methods and substrate materials for making microfluidic chips are limited, and the manufacturing technology requires high requirements and a long development cycle. Most of the microfluidic chips are still in the laboratory stage, so the technology development around the marketization of microfluidic chips still needs to be further explored.

Microfluidic chips detection is simple in terms of operation, is good in terms of accuracy, is intuitive in terms of results, is easy to store for a long time, involves less reagent consumption, is portable and can detect a large number of pathogenic microorganisms in a short period of time. The advantages bring broad prospects for the detection of airborne microorganisms. It can be used as an on-site laboratory for public health emergencies, greatly improving the detection efficiency and accuracy of public health emergencies and playing an important role in responding to public health emergencies. The structure of microfluidic chips is miniaturized and complex, the function is continuously increased, the scope of use is getting wider and wider and it has high application value in exploring the characteristics of microbiology. The use of microfluidic chips to detect airborne microorganisms is in the exploratory stage, and its application in clinical detection and environmental assessment detection is still very limited. At present, the manufacturing methods and materials of microfluidic chips are relatively limited, the manufacturing cost is relatively high and the detection types and detection limits need to be improved. These shortcomings make it less used in real life. With the development of various technologies, the combination of microfluidic chips with 3D printing, digital PCR, gene chips, flow cytometry and mass spectrometry detection technologies has produced more on-site instant detection products, which are applied to the safety analysis of airborne microorganisms. Then, it will provide help for the prevention and control of human environmental health.

## 6. Conclusions

Some epidemic diseases, such as influenza, Ebola, tuberculosis and pneumonia, are generally caused by the transmission of airborne microorganisms, especially pathogenic microorganisms, posing threats to human health and public safety.The detection and monitoring of air microorganisms are of great significance to disease prevention and control and environmental purification, especially the monitoring of densely populated places and the environment in hospitals, which is of vital importance to public safety.

Compared with traditional sampling and separation analysis methods, the main advantages of microfluidic chips in the detection of airborne microorganisms are: (1) Simple operation: The results are not easily disturbed. Integrating multiple experimental steps on a chip saves detection time, and at the same time, the closed structure ensures that the detection is not interfered with by the external environment. (2) High enrichment efficiency and low reagent consumption: By using the design of microchannels to increase the enrichment rate of airborne microorganisms, the required detection reagents are far less numerous than those required by traditional methods, which saves detection costs. (3) High efficiency: The microchannels in the chip can be set to independent multi-channels, and multiple tests can be performed on one sample at the same time, or multiple samples can be tested at the same time to ensure the accuracy and speed of the test results. (4) Portability: Integrated with the detection system, the volume is small, and continuous detection can be carried out, which is convenient for point-of-care detection.

## Figures and Tables

**Figure 1 micromachines-13-01576-f001:**
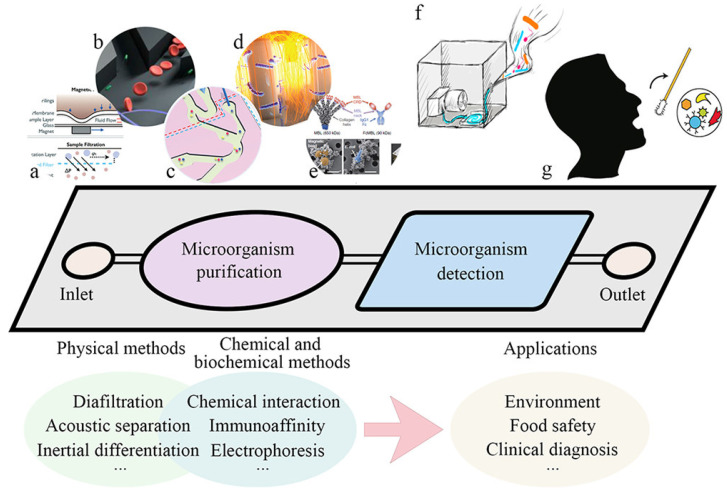
Schematic illustration of on-microchip microbial analysis, combining (**a**–**e**) on-chip purification of target microbes and detection methods for (**f**,**g**) applications, e.g., environmental monitoring and clinical diagnosis. Reproduced with permission from ref. [25]. Copyright 2018 American Chemical Society.

**Figure 2 micromachines-13-01576-f002:**
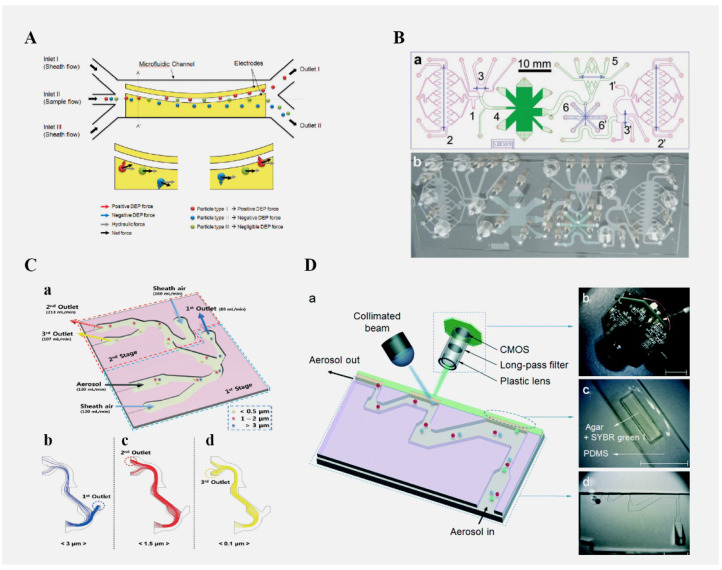
(**A**) Schematic diagram of the DEP separator. Reproduced with permission from ref. [89]. Copyright 2009 American Chemical Society. (**B**) Schematic drawing of laboratory-on-chip. (a) LOC-DLA, (b) picture of its assembly with manifold. Reproduced with permission from ref. [91]. (**C**) Schematic diagram of the inertial separation principle and the Dean vortices in the curved microchannel. (a) Design and operating conditions of the separator. (b), (c), (d) Trajectories of airborne particles 0.1, 1.5, 3 μm in diameter in the separator, respectively. Reproduced with permission from ref. [92]. Copyright 2015 Royal Society of Chemistry. (**D**) Structure of the centrifugation-based microfluidic chips for the collection of bioaerosols. (a) Schematic diagram of the present airborne microbe detection chip, (b) mini-fluorescent microscope, (c) disposable cartridge-type PDMS impaction plate with an impaction zone, and (d) microchannel with three impaction stages (all scale bars indicate 1 cm). Reproduced with permission from ref. [90]. Copyright 2014 Royal Society of Chemistry. Copyright 2011 Royal Society of Chemistry.

**Figure 3 micromachines-13-01576-f003:**
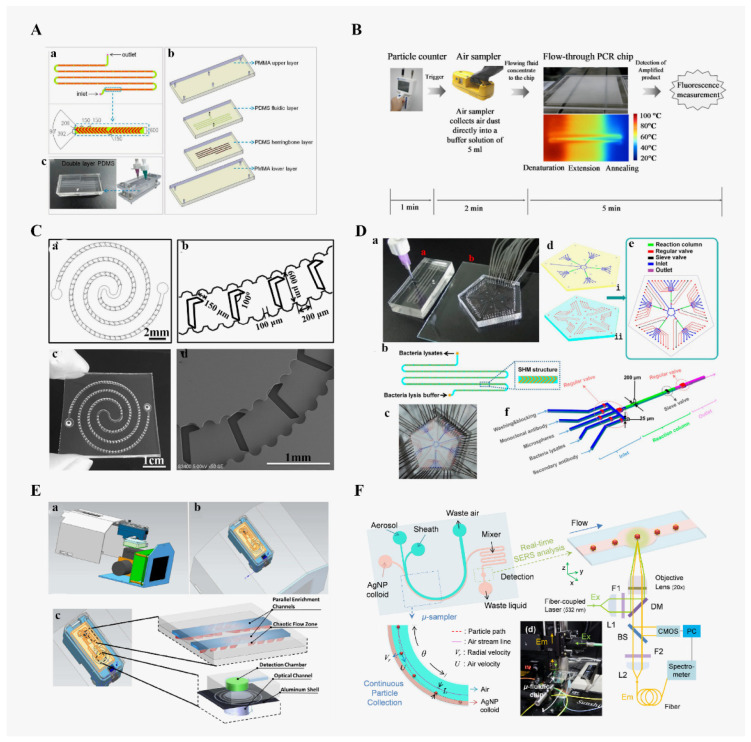
(**A**) Microfluidic chip with herringbone structures. (a) chip design and enlarged graph showing detailed structure of the microchannels (units, μm), (b)different components of the device, and (c) picture of the assembled chip and connectors. Reproduced with permission from ref. [94]. Copyright 2013 American Chemical Society. (**B**) Recommended combination for rapid field detection of biological agents. Reproduced with permission from ref. [96]. Copyright 2009 Elsevier B.V. (**C**) Double-spiral sawtooth wave-shaped microfluidic chip with herringbone structures. (a) AutoCAD model of the chip consisting of double spiral microchannels embedded with sawtooth wave-shaped and herringbone structures. (b) Magnified view of the spiral microchannel showing detailed design and dimensions. (c) Photograph of the PDMS microfluidic chip. (d) SEM image of the magnified spiral microchannel. Reproduced with permission from ref. [95]. Copyright 2016 American Chemical Society. (**D**) On-chip fluorescence immunoassay for detection of the bioaerosols containing *Mycobacterium tuberculosis.* (a) Image of the system for airborne bacteria rapid enrichment and bacteriological diagnosis. (b) Schematic illustration and detailed structure of enrichment microfluidic chip. (c) Image of the microfluidic immunoassay chip. (d,e) Schematic illustrations of designed immunoassay microfluidic chip. (f) Enlarged diagram showing detailed structure of microfluidic immunoassay chip. Reproduced with permission from ref. [77]. Copyright 2014 American Chemical Society. (**E**) 3D structure microfluidic chip with herringbone structures. (a) 3D structure of device, (b) An overhead view of microfluidic chip positioned in the aluminum shell (blue). (c) Detailed diagram of the key module. Reproduced with permission from ref. [97]. Copyright 2016 Royal Society of Chemistry. (**F**) Design of a continuous surface-enhanced Raman spectroscopy (SERS) system for detecting airborne biological particles. Reproduced with permission from ref. [98]. Copyright 2020 Elsevier B.V.

**Figure 4 micromachines-13-01576-f004:**
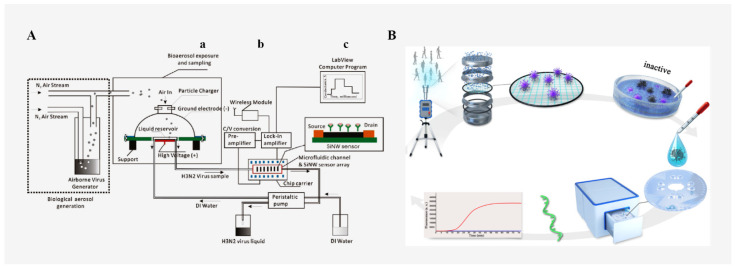
(**A**) Micro-optofluidic platform for the real-time, continuous detection of airborne microorganisms. (a) bioaerosol sampling and delivery, (b) antibody modified silicon nanowire based biosensor, and (c) signal amplification, detection, and online monitoring Reproduced with permission from ref. [101]. Copyright 2011 American Chemical Society. (**B**) Schematic diagram of the SARS-CoV-2 integrated sampling/monitoring microfluidic platform. Reproduced with permission from ref. [103]. Copyright 2021 American Chemical Society.

**Figure 5 micromachines-13-01576-f005:**
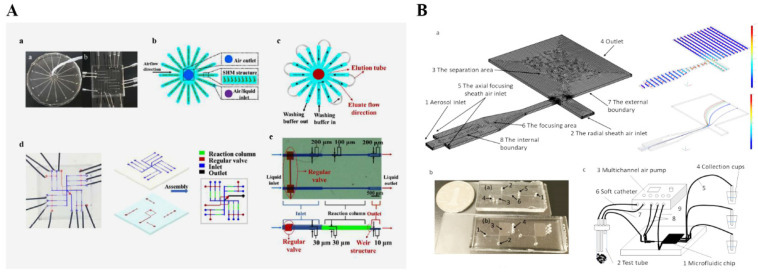
(**A**) On-chip fluorescence immunoassay for detection of the bioaerosols containing fungal spores. (a) airborne spore enrichment chip and detection chip. (b) Schematic illustration of the enrichment chip when sampling air. (c) Schematic illustration of the enrichment chip when eluting spores. (d) Image of the detection chip and schematic illustration of the detection chip. (e) Image and illustration of the detection units on the chip. Reproduced with permission from ref. [106]. Copyright 2018 American Chemical Society. (**B**) Microfluidic extraction chip simulation, physical and experimental device. (a) Simulation analysis, (b) the air spore extraction chip physical map, and (c) the air spore extraction experimental platform. Reproduced with permission from ref. [105]. Copyright 2019 American Institute of Physics.

## Data Availability

Not applicable.

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
