# Peer review of "Application of Microfluidic Chips in the Detection of Airborne Microorganisms"

_micromachines, 2022, doi:10.3390/mi13101576_

Round 1
Reviewer 1 Report
This manuscript discussed microfluidics and its application in airborne microorganisms detection. The manuscript is well organized, and the scope of this manuscript is matched well with the special issue of Micromachines. I therefore recommend that the manuscript be published after the following comments are taken into account.
Major comments:
(1) Collection of airborne microorganisms is the necessary prior step for detection, and microfluidics with special design (e.g., herringbone-based structure and centrifuge-based structure) have unique advantages over other conventional methods. However, the authors have not discussed this thoroughly.
(2) In this manuscript, the authors discussed three types of airborne microorganisms detection using microfluidics. Viruses, bacteria, and fungi are different sized microorganisms with different biological properties, these size and biological differences could significantly influence the design of microfluidic systems. Authors should provide insights on the design rules of microfluidic systems for each case based on their differences instead of simply listing the recent publication in each direction.
(3) The resolution of images is poor, especially Fig 3 and 4.
Minor comments:
(1) Some typos in the manuscripts, like “microflfluidic” in figure caption of Fig 1 and 2.
The last sentence of the abstract is weirdly arranged.
Author Response
Major comments:
- Collection of airborne microorganisms is the necessary prior step for detection, and microfluidics with special design (e.g., herringbone-based structure and centrifuge-based structure) have unique advantages over other conventional methods. However, the authors have not discussed this thoroughly.
A: Thank you for your advice. We have added some data and descriptions of the collection steps, such as Lines 316-317,335-337.
- In this manuscript, the authors discussed three types of airborne microorganism’s detection using microfluidics. Viruses, bacteria, and fungi are different sized microorganisms with different biological properties, these size and biological differences could significantly influence the design of microfluidic systems. Authors should provide insights on the design rules of microfluidic systems for each case based on their differences instead of simply listing the recent publication in each direction.
A: Thank you for your advice. We have added some specific microfluidic chips design content in 4.2 and 4.4.
(3) The resolution of images is poor, especially Fig 3 and 4.
A: We apologize for this. It was our mistake during figure preparation. During our high resolution of images preparation, the image was somehow shrank. We have corrected the images.
Minor comments:
(1) Some typos in the manuscripts, like “microflfluidic” in figure caption of Fig 1 and 2.
The last sentence of the abstract is weirdly arranged.
A: Thank you for your reminding. We have worked on the manuscript for a long time and the repeated addition and removal of sentences and sections obviously led to poor readability.

Reviewer 2 Report
The manuscript by Wang et al. provides a review of microfluidic chips that have been developed to detect airborne microorganisms. Although the authors reviewed and included many interesting microfluidic techniques for detecting airborne microorganisms which is very important and highly needed and appreciated with the ongoing COVID 19 pandemic, the report as it currently stands is not of sufficient quality to warrant publication. I have several comments that would require a major revision before acceptance.
The second use of the article “the” should be dropped off from the title: “Application of Microfluidic Chips in the Detection of the Airborne Microorganisms”
Nouns are missing after many words and statements throughout the article (e.g. after “of” in “the advances on the collection and detection of on microfluidic chips, after “microfluidic” in “and demonstrated a 5-layer microfluidic fabricated with it.”, etc.).
The following statement “for bioaerosol collection and detection were also discussed.” is added at the bottom of the abstract and is standing on its own. This statement should be either removed or added to the abstract or main text somewhere.
The manuscript should be properly revised to comprehensively improve the quality of the English being used for a scientific paper. There are many very clear and avoidable mistakes throughout the manuscript. Many words are capitalized in the middle of the sentence or misspelled, and some statements are incomplete and can’t stand on their own (e.g. “relatively difficult Control or prevention”, “microorganisms in aerosols Mainly depending”, “microflfluidic”, “within 9 minutes. 100%.”, “infections. A promising platform. In”, “(as shown in Figure 3B).including automatic”, etc.)
The reference section should be corrected to be consistent with the standard requirement of the journal Micromachines. Also, most of the references have extra letters in between brackets e.g. [M] or [J]
The authors are encouraged to give an example of a safety limit or reference for the following statement: “so that the level of pathogenic microorganisms in hospitals is maintained below the safety line to ensure patient health.”
The authors are encouraged to give examples of the “traditional analysis methods,” in “Compared with the traditional analysis methods, it has obvious innovation advantages.”
The following 2 sentences are confusing and somewhat contradicting “In recent years, several methods for bioaerosol collection and detection based on microfluidics have been developed and reported. However, there are relatively few reports on the collection and detection of airborne microorganisms using microfluidic chips.” The authors are encouraged to rewrite these two sentences to make their point clearer to the reader.
“Figure 1.” is a collection of figures with permission taken from respective authors to include in the original cited article “[20]”. Authors are encouraged check if they are allowed to use “Figure 1” in this article without first obtaining permission from the original authors. The same goes for the remaining figures in the article.
The authors are encouraged to add the following recently published advantages to the paper-based technology section after “good biocompatibility, and the ability to immobilize biological macromolecules.” to provide the reader with a more comprehensive review of recent advancements in microfluidic technology:
“Microfluidic Paper-based Analytical Devices can also make use of composite materials [a], valve technology [b], lightbox and cellphone technology [c], and optimized device architecture [d] to enhance detection performance.”
a. A. Charbaji, W. Smith, C. Anagnostopoulos, M. Faghri, Zinculose: A New Fibrous Material with Embedded Zinc Particles, Eng. Sci. Technol. an Int. J. 24 (2) (2021). https://doi.org/10.1016/j.jestch.2020.09.005
b. L.A. Ireta-Muñoz, I. Cueva-Perez, J.J. Saucedo-Dorantes, A. Pérez-Cruz, A Paper-Based Cantilever Beam Mini Actuator Using Hygro-Thermal Response. Actuators. 11, 94 (2022). https://doi.org/10.3390/act11030094
c. A. Carrio, C. Sampedro, J.L. Sanchez-Lopez, M. Pimienta, P. Campoy, Automated Low-Cost Smartphone-Based Lateral Flow Saliva Test Reader for Drugs-of-Abuse Detection. Sensors. 15 (2015). https://doi.org/10.3390/s151129569
d. A. Charbaji, H. Heidari-Bafroui, C. Anagnostopoulos, M. Faghri, A New Paper-Based Microfluidic Device for Improved Detection of Nitrate in Water, Sensors. 21 (2021). https://doi.org/10.3390/s21010102
The authors are required to clarify what they mean and add a reference to the following statement: “so PLA has good biocompatibility and has been adopted by the United States.”
The authors are required to provide a reference to this statement: “Approved by the Food and Drug Administration for its application to the human body,”
“(1)” is missing before “The solid impact method uses a sampler to…” in section 3.1
In section 4.1, the figure number is incorrect and the same is true for the remaining figures in the article as well. Also, the quality of this figure needs to be enhanced. Higher resolution images should be used. There’s a red wiggly line in “A”, the words are very pixelated in “B” and the aspect ratio isn’t properly maintained in “D”.
The authors should clarify their statement in “The enrichment efficiency can be approached within 9 minutes. 100%.”
Most of the words in the figure in section 4.2 are pixelated. The authors are encouraged to upload higher resolution pictures. Also, they are encouraged to make the pictures larger. The same goes for the figures in section 4.3 and 4.4.
The authors should add a “challenges and future trends” section between section 4 and the conclusion that summarizes the challenges faced using microfluidic technology in detecting airborne microorganisms and what they believe the future trend going forward will be.
The paragraph starting with “Microfluidic chip detection is simple in operation, … the prevention and control of human environmental health” doesn’t belong in the conclusion.
Finally, I recommend that the authors review this article to avoid the many grammatical mistakes found throughout the text. The author should also make the article easier to follow by the reader and include more details and information about the technologies that have already been developed and what they believe the future trend will be.

Author Response
The manuscript by Wang et al. provides a review of microfluidic chips that have been developed to detect airborne microorganisms. Although the authors reviewed and included many interesting microfluidic techniques for detecting airborne microorganisms which is very important and highly needed and appreciated with the ongoing COVID 19 pandemic, the report as it currently stands is not of sufficient quality to warrant publication. I have several comments that would require a major revision before acceptance.
The second use of the article “the” should be dropped off from the title: “Application of Microfluidic Chips in the Detection of the Airborne Microorganisms”
A: Thank you of your reminding. The second use of the article “the” has be dropped off from the title.
Nouns are missing after many words and statements throughout the article (e.g. after “of” in “the advances on the collection and detection of on microfluidic chips, after “microfluidic” in “and demonstrated a 5-layer microfluidic fabricated with it.”, etc.).
A: We are sorry for this mistake. We have examined the problem carefully.
The following statement “for bioaerosol collection and detection were also discussed.” is added at the bottom of the abstract and is standing on its own. This statement should be either removed or added to the abstract or main text somewhere.
A: We are sorry for this mistake. We have removed this sentence.
The manuscript should be properly revised to comprehensively improve the quality of the English being used for a scientific paper. There are many very clear and avoidable mistakes throughout the manuscript. Many words are capitalized in the middle of the sentence or misspelled, and some statements are incomplete and can’t stand on their own (e.g. “relatively difficult Control or prevention”, “microorganisms in aerosols Mainly depending”, “microflfluidic”, “within 9 minutes. 100%.”, “infections. A promising platform. In”, “(as shown in Figure 3B). including automatic”, etc.)
A: We apologize for the poor language of our manuscript. We have corrected and checked the mistakes one by one.
The reference section should be corrected to be consistent with the standard requirement of the journal Micromachines. Also, most of the references have extra letters in between brackets e.g. [M] or [J]
A: The references have been corrected one by one, and the extra letters [M] or [J] in between brackets have been removed.
The authors are encouraged to give an example of a safety limit or reference for the following statement: “so that the level of pathogenic microorganisms in hospitals is maintained below the safety line to ensure patient health.”
A: Some sentences have been added in the paper, in lines 70-73: “About 10% of patients will be affected by nosocomial microbial infection. Pathogenic fungi in hospitals such as Candida and Aspergillus fumigatus may cause pneumonia. Pneumonia infection is not common in patients with normal immune function, but can cause the death of organ transplant recipients or immunocompromised patients [10,11]. Protective equipment removal and patient rooms had high concentrations per titer of SARS-CoV-2 (varying from 0.9 × 103 to 4.0 × 103 copies/m3 and 3.8 × 103 to 7.2 × 103 TCID50/m3) in hospital settings [14].”.
The authors are encouraged to give examples of the “traditional analysis methods,” in “Compared with the traditional analysis methods, it has obvious innovation advantages.”
A: Thank you for your advice. “traditional analysis methods, such as culture, microscopy and counting method “has been added in the sentence, in line 83.
The following 2 sentences are confusing and somewhat contradicting “In recent years, several methods for bioaerosol collection and detection based on microfluidics have been developed and reported. However, there are relatively few reports on the collection and detection of airborne microorganisms using microfluidic chips.” The authors are encouraged to rewrite these two sentences to make their point clearer to the reader.
A: The sentence has been amended to “In recent years, several equipment systems based on microfluidics have been developed and reported for collection and detection of microorganisms.” in lines 85-87.
“Figure 1.” is a collection of figures with permission taken from respective authors to include in the original cited article “[20]”. Authors are encouraged check if they are allowed to use “Figure 1” in this article without first obtaining permission from the original authors. The same goes for the remaining figures in the article.
A:Thanks for reminding.
The authors are encouraged to add the following recently published advantages to the paper-based technology section after “good biocompatibility, and the ability to immobilize biological macromolecules.” to provide the reader with a more comprehensive review of recent advancements in microfluidic technology:
“Microfluidic Paper-based Analytical Devices can also make use of composite materials [a], valve technology [b], lightbox and cellphone technology [c], and optimized device architecture [d] to enhance detection performance.”
- Charbaji, W. Smith, C. Anagnostopoulos, M. Faghri, Zinculose: A New Fibrous Material with Embedded Zinc Particles, Eng. Sci. Technol. an Int. J. 24 (2) (2021). https://doi.org/10.1016/j.jestch.2020.09.005
- A. Ireta-Muñoz, I. Cueva-Perez, J.J. Saucedo-Dorantes, A. Pérez-Cruz, A Paper-Based Cantilever Beam Mini Actuator Using Hygro-Thermal Response. Actuators. 11, 94 (2022). https://doi.org/10.3390/act11030094
- Carrio, C. Sampedro, J.L. Sanchez-Lopez, M. Pimienta, P. Campoy, Automated Low-Cost Smartphone-Based Lateral Flow Saliva Test Reader for Drugs-of-Abuse Detection. Sensors. 15 (2015). https://doi.org/10.3390/s151129569
- Charbaji, H. Heidari-Bafroui, C. Anagnostopoulos, M. Faghri, A New Paper-Based Microfluidic Device for Improved Detection of Nitrate in Water, Sensors. 21 (2021). https://doi.org/10.3390/s21010102
A: Some content“In recent years, paper chips have been widely used in food safety, environmental residual pollution, pathogen detection and other fields [47]. Research on paper chips in infectious disease and pathogen detection shows priority, and paper chips are well suited for point-of-care detection of viruses, especially in poor areas [48].”has already been added to the paper in lines 157-160.
The authors are required to clarify what they mean and add a reference to the following statement: “so PLA has good biocompatibility and has been adopted by the United States.”
A: Polylactic acid (PLA) is a common biodegradable thermoplastic that can be used as a material for 3D-printed microfluidic chips. In addition, the reference [49] has been added.
The authors are required to provide a reference to this statement: “Approved by the Food and Drug Administration for its application to the human body,”
A:The reference [50] has been added.
“(1)” is missing before “The solid impact method uses a sampler to…” in section 3.1
A: Thanks for your reminding. ‘(1)’ has been added before “The solid impact method uses a sampler to…”
In section 4.1, the figure number is incorrect and the same is true for the remaining figures in the article as well. Also, the quality of this figure needs to be enhanced. Higher resolution images should be used. There’s a red wiggly line in “A”, the words are very pixelated in “B” and the aspect ratio isn’t properly maintained in “D”.
A: We have carefully modified the figures.
The authors should clarify their statement in “The enrichment efficiency can be approached within 9 minutes. 100%.”
A: This sentence has been rewritten:“The enrichment efficiency can be approached 100% within 9 minutes.” In lines:359-360.
Most of the words in the figure in section 4.2 are pixelated. The authors are encouraged to upload higher resolution pictures. Also, they are encouraged to make the pictures larger. The same goes for the figures in section 4.3 and 4.4.
A: The figures have been improved.
The authors should add a “challenges and future trends” section between section 4 and the conclusion that summarizes the challenges faced using microfluidic technology in detecting airborne microorganisms and what they believe the future trend going forward will be.
A: “challenges and future trends” section has been added.
The paragraph starting with “Microfluidic chip detection is simple in operation, … the prevention and control of human environmental health” doesn’t belong in the conclusion.
A: This paragraph has been revised and moved to “challenges and future trends” section.
Finally, I recommend that the authors review this article to avoid the many grammatical mistakes found throughout the text. The author should also make the article easier to follow by the reader and include more details and information about the technologies that have already been developed and what they believ

Reviewer 3 Report
Comments for “Application of Microfluidic Chips in the Detection of the Airborne Microorganisms” (micromachines-1759941)
In this review, the authors summarized microfluidic chips for detection of airborne microorganisms. The topic is interesting, and will gain wide readership. Generally, microfluidic chips for microorganism detection are carried out in solution-based reactions. So, the main challenge for airborne microorganism detection on chip should be given. From this reviewer, the collection step is key for subsequent analysis, so more data and descriptions are needed. I also suggest authors to discuss paper-based microfluidics for airborne microorganism detection. Besides, English grammar and language check throughout the manuscript are necessary. Examples are as follows.
For the title, Microorganisms or Microorganism, please check
“Here we summarize the advances on the collection and detection of based on microfluidic chips.”
“for bioaerosol collection and detection were also discussed.”
“such as: active”
“it is helped to”
“difficult Control”
Author Response
“Application of Microfluidic Chips in the Detection of the Airborne Microorganisms” (micromachines-1759941)
In this review, the authors summarized microfluidic chips for detection of airborne microorganisms. The topic is interesting, and will gain wide readership. Generally, microfluidic chips for microorganism detection are carried out in solution-based reactions. So, the main challenge for airborne microorganism detection on chip should be given. From this reviewer, the collection step is key for subsequent analysis, so more data and descriptions are needed. I also suggest authors to discuss paper-based microfluidics for airborne microorganism detection. Besides, English grammar and language check throughout the manuscript are necessary. Examples are as follows.
For the title, Microorganisms or Microorganism, please check
“Here we summarize the advances on the collection and detection of based on microfluidic chips.”
“for bioaerosol collection and detection were also discussed.”
“such as: active”
“it is helped to”
“difficult Control”
A: We apologize for the poor language of our manuscript. We will be happy to edit the text further, based on helpful comments from the reviewers. We worked on the manuscript for a long time and the repeated addition and removal of sentences and sections obviously led to poor readability. We really hope that the flow and language level have been substantially improved.
We have added some data and descriptions of the collection steps, such as Lines 316-317,335-337.We have added some content about paper-based microfluidic chips. In addition, the picture has been modified.

Round 2
Reviewer 2 Report
The authors have improved the manuscript according to initial comments and recommendations provided earlier. However, the authors are required to address the following recommendations:
Many grammatical mistakes are still present throughout the article, and the authors are required to proofread the manuscript again to avoid them (e.g. lines 22-23 “Besides, the challenges and trends”, line 34 “it is helped to”, line 50 “relatively difficult control or prevention”, etc.)
In line 18, a better word than “hotspot” should be used.
In line 44, the font of “10μm” is inconsistent.
In line 60, the following sentence can’t stand on its own: “Infectious diseases, respiratory diseases and even tumors.” The authors would also need to provide a reference for the tumors part of the sentence.
In lines 68-69, the authors are required to rewrite the below sentence to make their point clearer to the reader “Therefore, hospitals should set up a more indoor environment than normal.”
The statements in lines 63-66 and lines 69-72 are almost identical word for word.
In line 79, a better word than “imminent” should be used.
The 2 sentences in lines 86-89 have to be removed since they are contradicting and conflicting and in the cover letter, the authors mention that the idea has been conveyed in lines 85-87 “In recent years, several equipment systems based on microfluidics have been developed and reported for collection and detection of microorganisms. However, there are relatively a few reports on the collection and detection of airborne microorganisms using microfluidic chips.”
In lines 98-99, the following sentence can’t stand on its own “Then,it has been applied to actual business or family life in various fields.”
In line 149, the following sentence can’t stand on its own “In 2007, a new type of microfluidic technology using filter paper as a test carrier.”
In line 203, the following sentence can’t stand on its own “A non-contact processing technology [64].”
In line 205, what does “SLAM” in “SLAM method” stand for?
In line 219, the word “silicon” should be used before (Si).
In line 468-470, the following statement is incorrect “Although microfluidic chips can greatly reduce the consumption of samples and reagents, because of its expensive production cost and complex production process, this technology cannot be well applied in actual testing.”
The statement in line 471 “technical requirements for operators are relatively high.” Is contradicting the statement in line 474 “Microfluidic chip detection is simple in operation”
In line 479, the authors have to rewrite the following statement “and playing an emergency role in handling public health emergencies.”
In line 487, the authors have to rewrite the following statement “applications in practical applications”
Author Response
Dear reviewer:
Thank you so much for positive and constructive comments to revise our manuscript (1759941) entitled “Application of Microfluidic Chips in the Detection of Airborne Microorganisms”. According to your comments, we have carefully revised this manuscript. The responses to your comments and the main corrections in this paper are as follows.
- Many grammatical mistakes are still present throughout the article, and the authors are required to proofread the manuscript again to avoid them (e.g. lines 22-23 “Besides, the challenges and trends”, line 34 “it is helped to”, line 50 “relatively difficult control or prevention”, etc.)
Response: Thanks for your comments.We have revised these sentences.(Please see lines 23-24: “Besides, the challenges and trends for detection of airborne microorganisms by microfluidic chips was also discussed.”; line 36: “which helps to understand...”; line 52: “relatively difficult to control or prevent” , etc.
- In line 18, a better word than “hotspot” should be used.
Response: We have replaced “hotspot” by “highlights”. Please see line 18.
- In line 44, the font of “10μm” is inconsistent.
Response: Thank for your suggestions. The mistake has been corrected, please see line 46.
- In line 60, the following sentence can’t stand on its own: “Infectious diseases, respiratory diseases and even tumors.” The authors would also need to provide a reference for the tumors part of the sentence.
Response: According to your comments, we have added a reference. Please see line 64.
- 5.In lines 68-69, the authors are required to rewrite the below sentence to make their point clearer to the reader “Therefore, hospitals should set up a more indoor environment than normal.”
Response: We have revised this sentence. The improved sentence is “Therefore, hospitals have stricter requirements for microorganisms in indoor environments than in ordinary cases.”. Please see lines 74-75.
- The statements in lines 63-66 and lines 69-72 are almost identical word for word.
Response: We have corrected this duplication.
- 7.In line 79, a better word than “imminent” should be used.
Response:We have replaced “imminent” by “extremely urgent”. Please see line 80.
- 8.The 2 sentences in lines 86-89 have to be removed since they are contradicting and conflicting and in the cover letter, the authors mention that the idea has been conveyed in lines 85-87 “In recent years, several equipment systems based on microfluidics have been developed and reported for collection and detection of microorganisms. However, there are relatively a few reports on the collection and detection of airborne microorganisms using microfluidic chips.”
Response: Thanks for your comments.We have removed the sentence “ However, there are relatively a few reports on the collection and detection of airborne microorganisms using microfluidic chips. ”
- In lines 98-99, the following sentence can’t stand on its own “Then,it has been applied to actual business or family life in various fields.”
Response: We have revised this sentence. The improved sentence is “Then, it has been applied to real life in some fields [17-19].”. Please see line 100.
- 10.In line 149, the following sentence can’t stand on its own “In 2007, a new type of microfluidic technology using filter paper as a test carrier.”
Response: We have revised this sentence. The improved sentence is “Microfluidic paper-based analytical devices (μPADs) are a new concept first proposed in 2007, a novel microfluidic technology using filter paper as a test carrier [42].”. Please see lines 151-152.
- In line 203, the following sentence can’t stand on its own “A non-contact processing technology [64].”
Response: We have revised this sentence. The improved sentence is “Laser ablation method is a non-contact processing technology that uses a mask or directly according to the computer design to control the position of the XY direction of the laser, and uses high-energy laser irradiation on the material to destroy the interaction between polymer molecules, so as to make different types of microstructure and channel [68].”. Please see lines 206-209.
- In line 205, what does “SLAM” in “SLAM method” stand for?
Response: Thank for your suggestions. we have added words “simultaneous localization and mapping”. Please see line 212.
- In line 219, the word “silicon” should be used before (Si).
Response: Thank for your suggestions. We have replaced “Si” by “silicon”.
Please see line 226.
- In line 468-470, the following statement is incorrect “Although microfluidic chips can greatly reduce the consumption of samples and reagents, because of its expensive production cost and complex production process, this technology cannot be well applied in actual testing.”
Response: We have revised this sentence. The improved sentence is “Although microfluidic chips can greatly reduce the consumption of samples and reagents, but the methods and substrate materials for making microfluidic chips are limited, and the manufacturing technology requires high requirements and a long development cycle.”. Please see lines 487-489.
- 15.The statement in line 471 “technical requirements for operators are relatively high.” Is contradicting the statement in line 474 “Microfluidic chip detection is simple in operation”
Response: We have revised this sentence. The improved sentence is “Although microfluidic chips can greatly reduce the consumption of samples and reagents, but the methods and substrate materials for making microfluidic chips are limited, and the manufacturing technology requires high requirements and a long development cycle.”. Please see lines 487-489.
- 16.In line 479, the authors have to rewrite the following statement “and playing an emergency role in handling public health emergencies.”
Response: We have revised this sentence. The improved sentence is “playing an important role in responding to public health emergencies.”. Please see lines 498-499.
- 17.In line 487, the authors have to rewrite the following statement “applications in practical applications”
Response: We have revised this sentence. The improved sentence is “These shortcomings make it less used in real life.”. Please see line 506.
